# Reliability and Validity of the Polish Version of the Esophageal-Atresia-Quality-of-Life Questionnaires to Assess Condition-Specific Quality of Life in Children and Adolescents Born with Esophageal Atresia

**DOI:** 10.3390/ijerph19138047

**Published:** 2022-06-30

**Authors:** Anna Rozensztrauch, Robert Śmigiel, Dariusz Patkowski, Sylwester Gerus, Magdalena Kłaniewska, Julia Hannah Quitmann, Michaela Dellenmark-Blom

**Affiliations:** 1Department of Nursing and Obstetrics, Division of Family and Pediatric Nursing, Wroclaw Medical University, 50-556 Wrocław, Poland; robert.smigiel@umw.edu.pl (R.Ś.); magdalena.klaniewska@student.umed.wroc.pl (M.K.); 2Department of Paediatric Surgery and Urology, Wrocław Medical University, Borowska 213, 50-556 Wrocław, Poland; dariusz.patkowski@umw.edu.pl (D.P.); sylwester.gerus@umw.edu.pl (S.G.); 3Department of Medical Psychology, University Medical Center Hamburg-Eppendorf, 20251 Hamburg, Germany; j.quitmann@uke.uni-hamburg.de; 4Department of Paediatrics, Institute of Clinical Sciences, University of Gothenburg Sweden, 405 30 Gothenburg, Sweden; michaela.m.blom@vgregion.se; 5Department of Pediatric Surgery, Queen Silvia Children’s Hospital, 416 50 Gothenburg, Sweden

**Keywords:** quality of life, esophageal atresia, child, patient-reported outcome

## Abstract

Aim: This study reports the reliability and validity of the Polish version of the Esophageal Atresia Quality of Life (EA-QOL) questionnaires, which were originally developed in Sweden and Germany. Methods: A total of 50 families of children (23 aged 2 to 7, and 27 aged 8 to 17) with EA/TEF (esophageal atresia/tracheoesophageal fistula) participated in the study. The development and validation of the Polish version of the EA-QOL involved forward-backward translation of the survey items following the guidelines for cross-cultural translation, cognitive debriefing and evaluation of psychometric properties, including assessment of internal and retest reliability, linguistic validity, content validity, known-group validity and convergent validity. The medical records of patients and standardized questionnaires were used to obtain clinical data. The level of significance was *p* < 0.05. Results: The Polish versions of the EA-QOL questionnaires demonstrated strong linguistic and content validity, are slightly discriminative for esophageal and respiratory problems, but do not show convergent validity with the PedsQL 4.0 generic core scales. In terms of reliability, the internal consistency of the subscale and total scale of Polish versions as measured by Cronbach’s alpha is good, and retest reliability is excellent. Conclusions: The Polish versions of the EA-QOL questionnaires meet most psychometric criteria that confirm the EA-QOL questionnaires’ reliability and validity. This study enables application of these questionnaires in future research among children with EA in Poland and participation in international multicenter studies focusing on advancing knowledge of condition-specific QOL in this population. Future cross-cultural research using larger sample sizes is still needed to better address the relationship between condition-specific and generic QOL, as well as the discriminative ability of the EA-QOL questionnaires.

## 1. Introduction

Esophageal atresia (EA), a rare congenital malformation, occurs in 2.4/10,000 births, and is characterized by a discontinuity of the esophagus, which may be accompanied by a pathological connection to the trachea, resulting in a tracheoesophageal fistula (TEF). Infants born with EA need reconstructive surgery to restore continuity of the esophagus. In most cases, a primary anastomosis can be performed during the first days of life [1]. Nowadays, due to increased experience in surgical management and neonatal intensive care, the survival rates of patients regularly exceed 90% [2]. Nevertheless, long-term complications after repair of EA, including dysphagia and gastroesophageal reflux (GER), as well as feeding difficulties such as choking, food retention and prolonged meals are still common in the children. Respiratory distress due to recurrent respiratory infections and coughing are also common in children with EA [3,4,5,6,7,8], as are coexisting anomalies, present in 50% of the population.

Although the current body of knowledge of congenital EA is constantly growing, the etiology and pathogenesis of this condition remain unknown, and scientists are working on new theories in this regard. Surgical techniques in EA reconstruction in infants and care directly before and after reconstructive surgery of EA have advanced. Furthermore, we are more familiar with the long-term problems children and their families face after being discharged from hospital. However, we still know very little about how having a child with such a severe congenital defect affects the quality of life of the family. Witt et al. [9] noted that prior to 2018, five out of seven studies focusing on the impact of EA on the family reported a tremendous burden on parents due to their child’s illness. Looking through the prism of medical problems, we tend to forget that behind a sick child stands a family that must deal with a burden that may be beyond their strength. Therefore, improving the quality of life (QOL) in these children, as well as reducing their morbidity after reconstruction of EA, have become major long-term goals. However, in order to be able to optimize QOL of patients and their families, information of both generic and condition-specific QOL among these children is needed.

The research on QOL is a valuable source of information and a significant supplement to the data obtained from clinical observations. QOL is a multidimensional construct with many subdimensions capturing subjective experiences, and including psychosocial, physical, school well-being and functioning, as well as satisfaction with life [10]. A child’s illness is the basis for the risk of a deterioration in his/her QOL, manifested as reduced mental and physical health. However, application of the right treatment and its proper course, as well as the appropriate living conditions of the patient, can contribute to an improvement in QOL [11]. QoL was defined by Felce and Perry [12] as “an overall general well-being that comprises objective descriptors and subjective evaluations of physical, material, social, and emotional well-being together with the extent of personal development and purposeful activity, all weighted by a personal set of values”, while HRQOL is defined by Ebrahim [13] as “those aspects of self-perceived wellbeing that are related to or affected by the presence of disease or treatment”. The current approach towards understanding health-related QOL (HRQOL) is holistic in nature, as the treatment process pertains not only to the illness itself and the related suffering, but also to the entire existence of the patient. The majority of studies focusing on QOL in children after EA repair have used generic instruments [14], which only allow for the assessment of particular elements included in the definition of QOL and can be used regardless of the type of disease or its presence [15]. The development of a condition-specific QOL questionnaire for children with EA could contribute to a better understanding of the patient’s life situation, provide healthcare facilities with a more accurate and reliable tool for patient evaluation and in the long-term help improve patient care. Together with a generic QOL questionnaire, such a questionnaire is likely to allow for a comprehensive assessment of the patient’s condition [16,17,18]. In recent years, a set of age-specific condition-specific QOL questionnaires for children and adolescents with EA were developed and these have shown sound psychometric performance in Sweden, Germany and Turkey [3,19,20]. The aim of the present study was to translate and evaluate the psychometric properties of the Polish versions of the EA-QOL questionnaires.

## 2. Methods

The study procedure, which was in line with FDA/ISPOR guidelines, included translation, cognitive debriefing and field testing of the translated Polish version of the EA-QOL questionnaires. The EA-QOL questionnaires include a set of age-specific questionnaires for two age groups: 2–7 years and 8–17 years. The questionnaire for children aged 2–7 years comprises 17 items and assesses the child’s QOL in three domains: eating—7 items, physical health and treatment—6 items, social isolation and stress—4 items. Due to the children’s age, this is a parent-proxy reported questionnaire. The questionnaire for 8–17-year-olds has a child-reported and a parent-proxy-reported form. The content of the items in these two versions is identical, the only difference being the use of the third person in the parent-reported questionnaire. The questionnaire for children aged 8–17 years comprises 24 items and assesses QOL in four domains: eating—8 items, social relationships—7 items, body perception and health—5 items, and wellbeing—4 items. Respondents answer the questions using a five-point Likert scale [3].

The research project was approved by the Bioethics Committee of Wroclaw Medical University, Poland (permission no. KB-636/2020). The study was carried out following the Declaration of Helsinki and Good Clinical Practice guidelines.

### 2.1. Participants and Settings

The study procedure was conducted from April 2020 to March 2021 in accordance with FDA/ISPOR guidelines and a standardized study protocol written by the instrument developer(s) [3]. The study group was selected based on nonprobability sampling and comprised families of children after repair of EA in accordance with the ICD-10 criteria who were aged 2–17 years at the time of the study. The children were patients at the Department of Pediatric Surgery at the University Clinical Hospital in Wroclaw, one of the leading centers for reconstructive surgery of EA in Poland. All possible study participants received oral and written study information and were informed that the participation in the study is voluntary and confidential.

Children aged ≥ 8 years provided self-reports and, additionally, the questionnaire was completed by their parents. Due to their young age, children aged 2 to 7 years were represented by their parents, who completed the questionnaire. The participating parent provided confirmation of being the child’s main care provider, permanent residence with the child and the lack of any diagnosed mental illness. The exclusion criteria for study participation were a lack of written consent to participate in the study and child age < 2 years.

### 2.2. Clinical and Sociodemograhphic Data of the Patients

The patients’ clinical data were obtained from their medical records. These mainly included birth parameters, concomitant congenital disorders, Gross EA subtypes, postoperative complications and other surgical interventions. Health-related data at follow-up were collected using a structured questionnaire completed by the parents, including digestive and respiratory symptoms, and feeding difficulties of the child the last four weeks.

### 2.3. Translation and Linguistic Validation

The translation and cultural adaptation procedure was performed in accordance with the international standards described in the literature [21,22] and the study protocol guidelines provided by the instrument developer, which also provided a careful description of the aim of each item [3]. The EA-QOL questionnaires were forward-translated from Swedish into Polish by two independent translators, who were native Polish speakers and fluent in Swedish. The Polish versions were then verified and corrected by an expert—an individual proficient in Swedish, with expertise in providing care to children with EA. At this stage, a panel of experts, including a physician, a nurse, a physiotherapist, a psychologist and an expert in research on QOL in children with rare diseases, introduced a number of significant linguistic modifications stemming from the discrepancy of meaning of several questionnaire items. The resulting reconciled versions of the EA-QOL questionnaires were then sent to the author of the original instrument (MDB), where these were back-translated from Polish to Swedish by a native professional Swedish speaker fluent in Polish. This back-translation was reviewed for accuracy compared to the original Swedish version of the EA-QOL questionnaires and discussed in the group. As the elements included in the instrument did not deviate from the source version, and the translated Polish version of the EA-QOL questionnaire was considered to achieve linguistic validity, they were submitted for cognitive debriefing among the target population with the same graphic layout as the original.

### 2.4. Cognitive Debriefing

Documenting the target population input in item generation, along with evaluating patient understanding through cognitive interviewing, can provide the evidence for content validity [23]. The cognitive debriefing was conducted with 18 families. Medical records were reviewed for child clinical data. In line with the study protocol, the participants represented different severity levels of EA: severe EA clinical significant dysphagia, clinically significant gastro–esophageal reflux disease, received dilatation of esophagus, airway disease (three patients, for 2–7 years old and 8–17 years old); moderate EA, clinically significant dysphagia, gastro–esophageal reflux disease, received dilatation of esophagus or clinically significant airway disease with associated anomaly (four patients, for 2–7 years old and 8–17 years old); mild EA, dysphagia or gastroesophageal reflux disease or airway disease, no associated anomaly (two patients, for 2–7 years old and 8–17 years old).

The goal of the cognitive debriefing process was to identify any difficult or ambiguous items in the questionnaire and to determine whether item interpretation differed among the respondents compared to the instrument developers’ intentions of the items. The duration of the cognitive debriefing face-to-face interviews ranged between 30 min and 45 min. Additionally, at the end of each interview, the respondents were asked about any missing subjects related to the categories addressed in the questionnaire. Next, the results of cognitive debriefing were discussed among the researchers and the instrument developer. As there were no objections, we proceeded to conducting a field test.

### 2.5. Field Test

The aim of the field test was to evaluate reliability (internal reliability, retest reliability) and validity (known-groups validity, convergent validity). The psychometric assessment of the EA-QOL questionnaires was conducted with the use of a previously validated generic HRQOL questionnaire. This was the Pediatric Quality of Life (PedsQL) 4.0 questionnaire, which is composed of 23 items and assesses QOL within the preceding month in pre-school children (aged 5–7 years), school children (aged 8–12 years) and adolescents (aged 13–18 years). The report for children aged 2–4 years comprises 21 items and does not include school functioning and communication scales. Respondents rate the items on a five-point Likert scale, where: 0 denotes “never” and 4 “almost always” [24,25,26,27].

The EA-QOL-Questionnaire Field Test procedure

The psychometric evaluation procedure consisted of five steps:Printing the questionnaires and preparing pre-stamped envelopes;Recruiting patients and parents;Data collection and data registration in Excel/SPSS (including reminders to increase the response rate plus a retest study with a maximum time interval of three weeks between first and second measurement points);Data analysis;Agreement on the final EA-QOL questionnaires.

### 2.6. Statistical Analysis

The questionnaires’ descriptors used were the mean, median, SD and maximum and minimum values. The item responses of the EA-QOL questionnaires and PedsQL 4.0 were linearly transformed to a scale of 0–100, with higher scores reflecting better HRQOL. Statistical analysis was performed using Statistica 13 (Tibco Inc., Palo Alto, CA, USA). The internal reliability of subscale and total scores was confirmed if Cronbach’s alpha for the scales exceeded 0.7. Retest reliability was calculated using intra-class correlation coefficient (ICC). We expected moderate (0.5–0.75), good (0.75–0.9) and excellent (>0.90) ICC levels (36). Qualitative variables between groups were compared using the chi-squared test. Significant differences between the EA-QOL scores of the first and second measurement occasion were calculated using the Wilcoxon signed-rank test. Known-groups validity was tested for clinical subgroups (≥5 or more observations in each group) expected to differ in EA-QOL total scores, with the Mann-Whitney U test, by comparing EA-QOL total scores between EA children with or without esophageal, feeding and respiratory symptoms, respectively. Cohen’s d was used for a standardized interpretation; effect size > 0.2, small; >0.5, moderate; and >0.8, large. Convergent validity was examined using the Spearman’s rho (rs) correlation coefficient between the total scores of the EA-QOL questionnaires and of those of the PedsQL 4.0, expecting a correlation of at least ≥0.4. The level of significance in all tests was *p* < 0.05.

## 3. Results

### 3.1. Cognitive Debriefing

#### Content Validity

The samples participating in the cognitive debriefing were: clinically significant dysphagia, clinically significant gastro–esophageal reflux disease, received dilatation of esophagus, airway disease (three patients, for 2–7 years old and 8–17 years old); moderate EA—clinically significant dysphagia, gastro–esophageal reflux disease, received dilatation of esophagus or clinically significant airway disease with associated anomaly (four patients, for 2–7 years old and 8–17 years old); mild EA—dysphagia or gastroesophageal reflux disease or airway disease, no associated anomaly (two patients, for 2–7 years old and 8–17 years old). All respondents participating in the cognitive debriefing of the Polish version of the EA-QOL questionnaires for children 2–7 and for children 8–17 years stated that the items were relevant to their experience, easy to understand and were not sensitive to answers, suggesting strong content validity. The vast majority (89%) of participants reported a positive overall perception of the EA-QOL questionnaires.

### 3.2. Field Test

#### 3.2.1. Study Participants

Altogether, 50 families responded to the EA-QOL-questionnaires, including 23 children aged 2 to 7 years, and 27 children aged 8 to 17 years. The study population is presented in Table 1.

Children aged 2–7 more commonly presented with vomiting during or after meals (*p* = 0.04), signs of difficulty in swallowing food (*p* = 0.001) and airway infections (*p* = 0.001) compared to children with EA aged 8–17 years (Table 2).

#### 3.2.2. Descriptive Statistics, Internal Consistency and Retest Reliability

Descriptive statistics, internal consistency, and retest reliability of the age-specific EA-QOL questionnaires are shown in Table 3.

Internal consistency achieved acceptable standards (Cronbach’s alpha ≥ 0.7) in all domains on the EA-QOL questionnaire for children aged 2–7, all but one (body perception, 0.65) on the child-report version for children 8 to 17 years old and all but one (eating, 0.68) on the parent-reported version for children 8 to 17 years old.

In the case of the EA-QOL questionnaire for children aged 2–7 years, high levels of retest reliability were observed for the following scales: eating (ICC, 1.00), physical health and treatment (ICC, 0.95), social isolation and stress (ICC, 0.98), and total scales (ICC, 0.98). As for the EA-QOL questionnaire for children aged 8–17 years, high levels of retest reliability were observed for the following scales: eating (ICC, 1.00), social relationships (ICC, 1.00), body perception (ICC, 1.00), health and well-being (ICC, 1.00), and total scales (ICC, 1.00). As regards the EA-QOL questionnaire for the parents of children aged 8–17 years, high levels of retest reliability were observed for the following scales: eating (ICC, 1.00), social relationships (ICC, 0.98), body perception (ICC, 0.95), health and well-being (ICC, 1.00), and total scales (ICC, 0.99).

Table 4 presents the comparison of the EA-QOL questionnaire scores in the “test” and “retest” study. There were no statistically significant differences between the results. The test-retest showed mainly excellent results.

#### 3.2.3. Known-Groups Validity

Table 5 and Table 6 show the comparison of the total scores on the EA-QOL questionnaires between subgroups in children aged 2–7 and children aged 8–17 with and without digestive, respiratory and feeding symptoms, respectively. Due to the low sample size (5 < observations in a subgroup), several comparisons were statistically not feasible. In children with EA aged 2–7 years, the need for a long time to eat a meal (>30 min) was significantly associated with lower EA-QOL total scores (*p* = 0.032). In children with EA aged 8–17 years, lower EA-QOL total scores were significantly associated with feeding difficulties (*p* = 0.039–parent-report; *p* = 0.037–child-report), dyspnea at rest (*p* = 0.039) and physical activity (*p* = 0.039). All significant results indicated large effect sizes (>0.8). Although several results demonstrated lower EA-QOL total scores in the group of children aged 2–7 and 8–17 years with presence of several reported symptoms, they did mostly not reach statistical significance, *p* < 0.05.

#### 3.2.4. Convergent Validity

The EA-QOL total scores for children aged 2–7 years did not show a statistically significant correlation (*n* = 23, rs = 0.17) with the PedsQL 4.0 total scores. The EA-QOL total scores for children aged 8–17 years did not demonstrate a statistically significant correlation (child-report, *n* = 27, rs = 0.22; parent-report, *n* = 27, rs = 0.03) with the PedsQL 4.0 total scores. This suggests that convergent validity was not achieved, and that condition-specific and generic HRQOL in this study sample reflect different concepts.

## 4. Discussion

This study on the Polish versions of the EA-QOL questionnaires is the first of its kind in Poland and a milestone for research on children with EA, since it will enable a QOL assessment in research in Poland using an accurate and reliable condition-specific instrument for children with EA. The Polish versions of the EA-QOL questionnaires demonstrated satisfactory internal and retest reliability, strong linguistic and content validity and are slightly discriminative for esophageal, respiratory or feeding difficulties. The Polish versions of the EA-QOL questionnaires have the potential to further increase the knowledge of any risk factors of QOL impairments in areas with proven importance to the EA patients, such as eating, airway problems and aspects of social life. Hence, it will be possible to learn more about the specific problems of the disease that patients may be dealing with in everyday life and offer a great opportunity to improve the quality of care provided to these patients in the entire country.

The results of the latest research show that from the patient’s perspective of general well-being, the capacity for active daily functioning, the ability to participate in social roles, health satisfaction, and physical and mental performance are more important and decisive in determining the patient’s adherence to therapeutic recommendations [28,29].

As for reports on QOL in children after EA repair, it is clearly visible that research methodology is of key importance [21,30,31]. A literature review has shown various results, but overall suggests that total generic QOL scores are lowered in patients with EA [21]. However, looking at the research methods used, one cannot draw any definitive conclusions pertaining to the impact of EA on the patients’ QOL. Thus, the main argument justifying the need for the development of the EA-QOL questionnaire was the lack of a properly constructed condition-specific assessment to capture issues of importance to children and adolescents with EA and developed using international guidelines of patient-reported outcomes.

### 4.1. Reliability

The internal reliability of the Polish version of the EA-QOL questionnaires has been verified with Cronbach’s alpha, which is a measure of internal consistency of a given research instrument, showing that the Polish version of the EA-QOL questionnaire is mainly acceptable to good criteria [32], with subscale and total scale values referencing Cronbach’s alpha between 0.7 and 0.9 [33,34]. Compared to previous research, satisfactory internal reliability has also been found in the Swedish-German and Turkish versions of the EA-QOL questionnaires [3,20]. Moreover, in the Polish version of the EA-QOL questionnaires for children aged 2–7 and 8–17 there were no statistically significant differences between the test and retest results. The excellent retest results could be explained by the high response rate, which is similar to the Turkish version [20]. Chinapaw et al. [35] recommended that adequate time between test and retest has an influence on reliability. Bobakova et al. [36] suggested that if the time between the two questionnaires is short, the respondents might remember their answers. Taking the above into account, while working in accordance with the study protocol, we tried to avoid errors related to respondents remembering answers by using the time of 3 weeks between the test and retest.

### 4.2. Validity

This study confirmed strong content validity of the Polish versions of the EA-QOL questionnaires as defined by current standards [37]. All respondents participating in the cognitive debriefing of the Polish version of the EA-QOL questionnaires confirmed the relevance, clarity and adequacy of all the items. Firstly, a good translation may help prevent poorly translated instruments that threaten the validity of the research data. We complied with international standard to achieve linguistic validity of the Polish translation of the EA-QOL questionnaires, which may help to explain the study findings. Secondly, it its known that rare conditions pose specific challenges to psychometric evaluations of instruments, especially due to low sample size and heterogeneity within the condition [38]. Therefore, a cross-cultural approach in the development of HRQOL measurements strengthens the generalizability of the study findings and may help to explain the strong content validity of the Polish version of the EA-QOL questionnaires, which was found in this study. The EA-QOL questionnaires were developed in Sweden and Germany, which are two North-European countries with both similarities and differences with regard to healthcare system, patient advocacy groups and follow-up care. For example, pediatric surgical care in Sweden is centralized, while in Germany it is decentralized. However, both study centers provide long-term follow-up care for children with EA. The cross-cultural approach may also limit the challenges of the clinical heterogeneity of a condition [37].

It is desirable that condition-specific instruments can identify clinical parameters that are associated with lower QOL in children, e.g., to discriminatively identify individuals with a larger or lesser QOL burden [14]. The Polish versions of the EA-QOL questionnaires for children aged 2–7 and 8–17 seem only slightly discriminative. In the questionnaire version for children with EA aged 8–17 years, feeding difficulties, dyspnea at rest and physical activity significantly were associated with lower QOL. In comparison, both the Swedish-German and Turkish evaluation [3,20] showed good discriminative ability with regard to digestive morbidity in the version for the EA-QOL questionnaires for children 8–17 years, reflecting that esophageal symptoms and feeding difficulties are associated with reduced EA-QOL. However, many symptom groups were too small to be included in a reliable statistical analysis and the groups included in the investigation were still small, and smaller compared to the Swedish-German (*n* = 124) and Turkish (*n* = 105) field tests. Furthermore, comparing the symptom prevalence reported in the Swedish-German sample of the field test [3], the symptom prevalence seems mostly less in the Polish study sample. For example, in the 2–7 age group, the Swedish-German sample vs. the Polish sample, wheezing was reported in 43% vs. 17%, chest tightness in 24% vs. 4% and heartburn in 31% vs. 25% of respondents. In the 8–17 age group, the Swedish-German sample vs. Polish sample showed that 37% vs. 22% of respondents reported heartburn, 33% vs. 18% signs of difficulty swallowing, 18% vs. 7% vomiting problems and 12% vs. 3% chest tightness. Although the comparison has not been statistically tested, and the reasons for the variations are not known, the clinical characteristics may illustrate the challenge of heterogeneity of a rare condition such as EA. It is likely that these clinical characteristics may influence the evaluation of clinical known-group validity and contribute to our study findings.

Interestingly, the EA-QOL questionnaires for children aged 2–7 and 8–17 years did not show convergent validity in relation to the PedsQL 4.0 generic core scales, which is different to previous evaluations. A possible explanation could be the statistically low sample size. However, this may also be explained by condition-specific QOL for children with EA in Poland reflecting a different concept than generic QOL. Since the EA-QOL questionnaires achieved strong content validity, it would suggest important complementary information of the instruments, next to a generic QOL instrument.

### 4.3. Implications and Methodological Considerations

It has recently been pointed out that rare diseases of childhood impact not only on health but also on fundamental human rights. Children living with a rare disease have, according to United Nations of the convention on the rights of the child, the rights to healthcare and societal support to achieve optimal health and development [39]. The EA-QOL questionnaires were developed using focus groups with children and their parents to enable the child’s perspective and evaluation of his/her QOL [18]. Furthermore, the evaluation in Poland has confirmed its content validity, increasing the chance that the important needs of these children can be identified in future research and clinical care. QOL assessments have been increasingly used in the EU countries since treatment outcome evaluation based on the biological criterion became insufficient [40]. These studies comprise a valuable source of medical information, as they complement data obtained in the course of laboratory and diagnostic testing. The perspective they present differs from that seen through the prism of professional medical knowledge. The patient perceives their illness in the context of their own psycho-social situation and assesses their condition across all domains of life. Undoubtedly, such studies aim towards improving communication between medical staff and patients [41,42]. Mutual interactions within the treatment process necessitate a shift from the traditional approach focusing solely on the measures of physical health [43]. In fact, in a systematic review, it was found that integration of patient-reported outcome measurements increased the identification and discussion around HRQOL, especially in the psychosocial and emotional domains [44]. A previous study showed that the use of an assessment of HRQOL may promote insights about health and encourage children with chronic health conditions to discuss their outcomes with healthcare professionals [45]. In the context of EA, international follow-up guidelines provide no details, but underline the need for a QOL assessment in follow-up care [46]. Our study in 2019, the first and only such study conducted in Poland, confirmed that the assessment of the QOL in children with EA is an essential element of caring for children after repair of EA [47]. While the medical care for children with EA is well and systematically organized, there is not yet any procedure which allows us to conduct follow–up in terms of QOL. Although several evaluations of its applicability in a clinical context and for longitudinal assessment remain, the Polish psychometric evaluations pave the way for future use in research and clinical care. The Polish version of the EA-QOL research sample is small, but cooperation with other centers is being established. The study was conducted in a leading pediatric surgery center, so the group is clinically representative with a high response rate. However, the results point to the importance of cross-cultural international research, increasing the sample sizes and representativity of the study samples in children with EA.

## 5. Conclusions

As a source of stress and something difficult to comprehend and accept, a rare childhood disease risks having a negative effect on how a child and his/her family functions. The Polish version of the EA-QOL questionnaire meets most psychometric criteria that confirm the EA-QOL questionnaires’ reliability and validity and allow for identifying domains of life that pose problems to the patients and their family. This study enables application of the questionnaires into future research among children with EA in Poland and our participation in international multicenter studies focusing on advancing knowledge of condition-specific QOL in this population. Future cross-cultural research using larger sample sizes is still needed to better address the relationship between condition-specific and generic QOL as well as the discriminative ability of the EA-QOL questionnaires.

## Figures and Tables

**Table 1 ijerph-19-08047-t001:** Presentation of families responding to the EA-QOL-questionnaires for children aged 2 to 7 years (*n* = 23) and children aged 8 to 17 years (*n* = 27).

	Children 2–7 Years Old					Children 8–17 Years Old				
	N	%	Me	Min	Max	N	%	Me	Min	Max
Male	13	54.2				19	70.4			
Gestational age (in weeks)			37	30	41			37	30	41
Birth weight (in grams)			2360	1205	3370			2350	1010	3700
Multiple birth	11	45.8				1	3.7			
Associated anomalies										
Cardiovascular	9	37.5				5	18.5			
Anorectal	2	8.3				2	7.4			
Gastrointestinal excl. anorectal	4	16.7				2	7.4			
Uro-genital	2	8.3				6	22.2			
Limb	1	4.2				0	0.0			
Vertabral-skeletal	1	4.2				7	25.9			
Choanal atresia	0	0.0				0	0.0			
Eye	3	12.5				0	0.0			
Ear	1	4.2				1	3.7			
Central Nervous System	4	16.7				0	0.0			
Pulmonary	0	0.0				3	11.1			
Other	8	34.8				0	0.0			
VACTERL	1	4.2				3	11.1			
CHARGE	1	4.2				0	0.0			
Chromosomal abnormality	3	12.5				2	7.7			
Parental information										
Parental age (in years)			36	25	45			39	33	49
Cohabitant partner	22	91.7				23	85.2			
College/University	11	45.8				17	63.0			

N, number of participants; Me, median; Min, minimum; Max, maximum.

**Table 2 ijerph-19-08047-t002:** Follow-up characteristics of the group studied.

Variables		Children 2–7 Years Old (*n* = 23)					Children 8–17 Years Old (*n* = 27)					*p*-Value
		N	%	Me	Min	Max	N	%	Me	Min	Max	
Child age (years)				4.5	2.0	7.0			11.0	8.0	16.0	
Child weight (kg)				16.5	8.0	21.0			36.0	22.0	85.0	
Child height (cm)				110.0	75.0	128.0			150.0	108.0	175.0	
Number of siblings	0	7	29.2				10	37.0				
	1	13	54.2				13	48.1				
	2	2	8.3				3	11.1				
	3	2	8.3									
	8						1	3.7				
Heartburn		6	25.0				6	22.2				0.82 **
Vomiting during or after meals		7	29.2				2	7.4				0.04 **
Signs of difficulty in swallowing food		15	62.5				5	18.5				0.001 **
Food getting stuck		6	25.0				9	33.3				0.51 **
Complaints of pain while swallowing		3	12.5				0	0.0				0.06 **
Cough		15	62.5				13	48.1				0.30 **
Wheezing		4	16.7				5	18.5				0.86 **
Dyspnea at rest/physical activity		7	29.2				5	18.5				0.37 **
Chest tightness		1	4.2				1	3.7				0.93 **
Airway infections		16	66.7				6	22.2				0.001 **
Does your child have doctor-diagnosed asthma?		4	16.7				3	11.1				0.57 **

N, number of participants; Me, median; U Mann-Whitney; ** chi2 test.

**Table 3 ijerph-19-08047-t003:** Descriptive statistics, internal consistency, and external reliability of the EA-QOL-questionnaire for children aged 2 to 7 years (parent-report) and children aged 8 to 17 years (child- and parent-report).

EA-QOL Questionnaire Scores	Descriptive Statistics					Internal Reliability	External Reliability, Retest Study		
	Number of Items	Number of Respondents	Median	Min	Max	Cronbach’s Alpha	Number of Respondents	Level of Agreement, ICC	−95 CI; +95 CI
Children 2–7 years old (parent-report)									
Eating	7	23	67.9	39.3	96.4	0.70	23	1	1; 1
Physical health and treatment	6	23	62.5	16.7	95.8	0.87	23	0.95	0.90; 0.98
Social isolation and stress	4	23	62.5	0.0	100.0	0.80	23	0.98	0.97; 0.99
Total scores	17	23	62.5	25.8	92.3	0.70	23	0.98	0.96; 0.99
Children 8 to 17 years old (child-report)									
Eating	8	27	84.4	21.9	100.0	0.72	26	1	1; 1
Social relationships	7	27	96.4	53.6	100.0	0.82	26	1	1; 1
Body perception	5	27	100.0	65.0	100.0	0.65	26	1	1; 1
Health and well-being	4	27	87.5	62.5	100.0	0.75	26	1	1; 1
Total scores	24	27	90.6	61.3	100	0.84	26	1	1; 1
Children 8 to 17 years old (parent-report)									
Eating	8	27	81.3	21.9	100.0	0.68	26	1	1; 1
Social relationships	7	27	85.7	53.6	100.0	0.80	26	0.98	0.96; 0.99
Body perception	5	27	100.0	40.0	100.0	0.79	26	0.95	0.90; 0.98
Health and well-being	4	27	87.5	37.5	100.0	0.75	26	1	1; 1
Total scores	24	27	86.7	57.1	100.0	0.82	26	0.99	0.98; 1

**Table 4 ijerph-19-08047-t004:** Comparison of the EA-QOL questionnaire scores in the “test” and “retest” study.

	EA-QOL Questionnaire Scores										*p*-Value *
	Test					Retest					
	Mean	Me	Min	Max	SD	Mean	Me	Min	Max	SD	
Children 2–7 years old (parent-report)											
Eating	66.0	67.9	39.3	96.4	15.1	66.0	67.9	39.3	96.4	15.1	1.00
Physical health and treatment	60.5	62.5	16.7	95.8	21.8	57.5	62.5	16.7	87.5	23.3	0.70
Social isolation and stress	60.9	62.5	0.0	100.0	28.0	60.5	62.5	0.0	100.0	30.2	0.95
Total scores	62.5	64.4	25.8	92.3	16.2	61.3	62.5	25.8	92.3	17.7	0.90
Children 8 to 17 years old (child-report)											
Eating	81.6	84.4	21.9	100.0	15.9	81.5	82.8	21.9	100.0	16.2	0.99
Social relationships	85.4	96.4	53.6	100.0	18.3	84.3	91.1	53.6	100.0	18.1	0.68
Body perception	91.7	100.0	65.0	100.0	13.4	91.2	100.0	65.0	100.0	14.0	0.84
Health and well-being	86.6	87.5	62.5	100.0	13.3	86.1	87.5	62.5	100.0	13.3	0.88
Total scores	86.3	90.6	61.3	100.0	11.6	85.8	90.6	61.3	100.0	11.2	0.76
Children 8 to 17 years old (parent-report)											
Eating	78.9	81.3	21.9	100.0	16.7	78.9	81.3	21.9	100.0	16.7	1.00
Social relationships	83.7	85.7	53.6	100.0	16.5	83.7	85.7	53.6	100.0	16.5	1.00
Body perception	90.0	100.0	40.0	100.0	17.7	90.0	100.0	40.0	100.0	17.7	1.00
Health and well-being	84.0	87.5	37.5	100.0	17.2	84.0	87.5	37.5	100.0	17.2	1.00
Total scores	84.2	86.7	57.1	100.0	12.3	84.2	86.7	57.1	100.0	12.3	0.99

* Wilcoxon test.

**Table 5 ijerph-19-08047-t005:** Comparison of the total scores on the EA-QOL questionnaire between clinical subgroups in children aged 2–7. *p* < 0.05 values are significant.

	Children 2–7 Years Old (Parent-Report)							
	Yes			No			*p*-Value **	ES ***
	N *	Mean	SD	N *	Mean	SD		
Digestive symptoms								
Heartburn	6	60.4	20.1	17	63.2	15.3	1.000	0.16
Vomiting during or after meals	7	54.6	16.2	16	65.9	15.4	0.088	0.71
Signs of difficulty in swallowing food	14	62.9	14.7	9	61.8	19.2	0.705	−0.06
Food getting stuck	5	66.3	17.1	18	61.4	16.3	0.823	−0.29
Respiratory symptoms								
Cough	14	60.2	19.2	9	66.0	10.1	0.614	0.38
Dyspnea at rest/physical activity	6	53.7	16.6	17	65.5	15.4	0.151	0.74
Airway infections	15	65.0	16.4	8	57.6	15.6	0.245	−0.46
Feeding difficulties								
Avoids certain foods	19	63.5	15.1	4	57.4	22.7	0.776	−0.32
Eats a small portion	15	63.4	16.8	8	60.6	15.9	0.796	−0.17
Needs adjusted food consistency	13	59.4	14.8	10	66.4	17.9	0.278	0.43
Needs a long time to eat >30 min	13	56.3	15.3	10	70.5	14.2	0.032	0.96
Needs additional assistance while eating	14	59.1	15.7	9	67.6	16.4	0.244	0.53

EA = esophageal atresia; ES = effect size; OL = Quality of Life, * the number of patients included in the testing refers to patients with both available clinical data and EA-QOL total scores, ** U Mann-Whitney test, *** Cohen’s d used for a standardized interpretation; effect size > 0.2, small; >0.5, moderate; and >0.8, large.

**Table 6 ijerph-19-08047-t006:** Comparison of the total scores on the EA-QOL questionnaire between clinical subgroups in children aged 8–17. *p* < 0.05 values are significant.

	Children 8–17 Years Old (Parent-Report)								Children 8–17 Years Old (Child-Report)							
	Yes			No			*p*-Value	ES	Yes			No			*p*-Value	ES
	N	Mean	SD	N	Mean	SD			N	Mean	SD	N	Mean	SD		
Digestive symptoms																
Heartburn	6	80.9	13.8	21	85.1	12.1	0.466	0.32	6	82.1	15.0	21	87.5	10.6	0.321	0.42
Signs of difficulty in swallowing food	5	73.3	13.0	22	86.6	11.0	0.039	1.10	5	75.4	11.9	22	88.8	10.3	0.037	1.20
Food getting stuck	9	79.6	15.0	18	86.5	10.5	0.304	0.53	9	82.5	12.5	18	88.3	11.0	0.208	0.49
Respiratory symptoms																
Cough	13	83.0	12.0	14	85.3	13.0	0.482	0.18	13	83.8	12.7	14	88.6	10.5	0.320	0.41
Dyspnea at rest/physical activity	5	74.1	12.7	22	86.5	11.3	0.039	1.03	5	77.6	14.2	22	88.3	10.3	0.081	0.86
Airway infections	6	92.6	8.45	21	81.8	12.4	0.041	−1.02	6	93.0	8.45	21	84.4	11.8	0.066	−0.84
Feeding difficulties																
Avoids certain foods	6	79.7	15.5	21	85.4	11.4	0.448	0.42	6	83.4	14.3	21	87.2	11.0	0.620	0.30
Eats a small portion	8	82.8	9.9	19	84.7	13.4	0.396	0.16	8	79.1	13.2	19	89.4	9.7	0.041	0.89
Needs a long time to eat >30 min	8	88.6	11.7	19	82.3	12.4	0.184	−0.52	8	90.1	12.6	19	84.8	11.1	0.159	−0.45
Needs increased fluid intake to make it easier to swallow food	13	85.0	12.8	14	83.4	12.3	0.734	−0.13	13	84.0	14.5	14	88.5	8.0	0.771	0.38

## Data Availability

The data that support the findings of this study are available from the corresponding author, upon reasonable request.

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
