# Peer review of "Reliability and Validity of the Polish Version of the Esophageal-Atresia-Quality-of-Life Questionnaires to Assess Condition-Specific Quality of Life in Children and Adolescents Born with Esophageal Atresia"

_ijerph, 2022, doi:10.3390/ijerph19138047_

Round 1
Reviewer 1 Report
This paper considers significant problems such as reports on the reliability and validity of the Polish version of the Esophageal- Atresia- Quality- of -Life (EA-QOL) questionnaires.
Due to the usefulness and diagnostic effectiveness of the use of this questionnaire, it is very important to evaluate the Polish version.
This article provides a detailed step-by-step description of how it was validated. The research includes all stages of the correct assessment of the questionnaire. The results are presented in a legible and understandable way. I recommend this article for publication. A few comments and suggestions mentioned below must be improved before the manuscript is accepted for publication.
Specific comments:
Material and methods:
1) add information about the Bioethics Committee - at which institution the Committee is located.
2) due to the fact that the research concerns quite a specific group of participants. Were the group inclusion and exclusion criteria more detailed at the stage of recruiting for the research? If so, I would ask you to complete these criteria.
3) please review the article for grammar errors (spelling errors, double spaces).
Author Response
DEAR REVIEWER 2,
REVIEWER COMMENTS 1 This paper considers significant problems such as reports on the reliability and validity of the Polish version of the Esophageal- Atresia- Quality- of -Life (EA-QOL) questionnaires. Due to the usefulness and diagnostic effectiveness of the use of this questionnaire, it is very important to evaluate the Polish version. This article provides a detailed step-by-step description of how it was validated. The research includes all stages of the correct assessment of the questionnaire. The results are presented in a legible and understandable way. I recommend this article for publication. A few comments and suggestions mentioned below must be improved before the manuscript is accepted for publication.
Thank you for your encouraging remarks.
REVIEWER COMMENTS 2
Specific comments:
Material and methods:
1) add information about the Bioethics Committee - at which institution the Committee is located.
Thank you for this comment. Please see the following paragraph added in the result section
The research project was approved by the Bioethics Committee of Wroclaw Medical University, Poland (permission no. KB–636/2020). The study was carried out following the Declaration of Helsinki and Good Clinical Practice guidelines.
REVIEWER COMMENTS 3
2) due to the fact that the research concerns quite a specific group of participants. Were the group inclusion and exclusion criteria more detailed at the stage of recruiting for the research? If so, I would ask you to complete these criteria.
Due to the design of the study, the inclusion criterion were patients after correction of esophageal atresia in two studied age groups - 2-7 years old and 8- 17 years old. The only exclusion criterion from the study was the age of less than 2 years and the lack of consent to the study. The study concerned children after EA surgery, and only such children were taken into account. Additionally, we have added in the debriefing section the following paragraph:
Severe EA – clinical significant dysphagia, clinically significant gastro – esophageal reflux disease, received dilatation of esophagus, airway disease (3 patients, for 2-7 years old and 8-17 years old) ; moderate EA – clinically significant dysphagia, gastro – esophageal reflux disease, received dilatation of esophagus or clinically significant airway disease with associated anomaly (4 patients, for 2-7 years old and 8- 17 years old); mild EA – dysphagia or gastroesophageal reflux disease or airway disease, no associated anomaly (2 patients, for 2-7 years old and 8- 17 years old).
REVIEWER COMMENTS 3
3) please review the article for grammar errors (spelling errors, double spaces).
Thank you for this comment. Entire the article was corrected by a professional native Englishspeaker.
Reviewer 2 Report
Dear Authors,
I have read your valuable work with great interest and I have no suggestions to improve it. I would like to stress one limitation you mentioned at the end of your manuscript concerning a small research sample and encourage you to perform a multi-center study.
Author Response
Dear Reviewer,
Thank you very much for sending us the consensus opinion about requested revision of our manuscript entitled: Reliability, and Validity of the Polish Version of the Esophageal-Atresia-Quality-of-Life Questionnaires to Assess Condition-Specific Quality of Life in Children and Adolescents Born with Esophageal Atresia. We appreciate the thoughtful comments and we have modified the manuscript in response to your suggestions, which we believe will further improve its quality.
REVIEWER COMMENTS 1
I have read your valuable work with great interest and I have no suggestions to improve it. I would like to stress one limitation you mentioned at the end of your manuscript concerning a small research sample and encourage you to perform a multi-center study.
Thank you for your encouraging remarks.
Yes indeed. We have already established cooperation with specialized centers of pediatric surgery centers all over Poland, as well as in Europa, and we will carry out a multi-center study, increasing the study group.

Reviewer 3 Report
This study aims to assess the reliability and validity of Polish version of QoL questionnaire for patients/families with esophageal atresia.
I have a couple of comments:
Typo on line 96: weres à were
Line 111à “In line with the study protocol the participants represented different severity levels of EA: severe EA”
It seems that the sentence should continue?
Table 1: were there really 11 children (in age group 2–7 years) with multiple pregnancy (>45%) while only 1 child (3.7%) in older age group? Also, abbreviations are given for median (Me) and mean (M), but those are not used in the Table.
Table 2 is very confusing, not clear at all. Also, what does ‘y’ stand for which is used after symptoms like heartburn etc.
To summarize, the authors have done a great job with the translation and it appears to me that the questionnaire validation is carried out well. This is definitely relevant for the Polish population and especially families and children with esophageal atresia and the teams looking after them.
Author Response
Dear Reviewer,
Thank you very much for sending us the consensus opinion about requested revision of our manuscript entitled: Reliability, and Validity of the Polish Version of the Esophageal-Atresia-Quality-of-Life Questionnaires to Assess Condition-Specific Quality of Life in Children and Adolescents Born with Esophageal Atresia. We appreciate the thoughtful comments and we have modified the manuscript in response to your suggestions, which we believe will further improve its quality.
REVIEWER COMMENTS 1
This study aims to assess the reliability and validity of Polish version of QoL questionnaire for patients/families with esophageal atresia.
I have a couple of comments:
Typo on line 96: weres à were
Thank you for this comment. Entire the article was corrected by a professional native English-speaker.
REVIEWER COMMENTS 2
Line 111à “In line with the study protocol the participants represented different severity levels of EA: severe EA”
It seems that the sentence should continue?
Thank you for this comment. Please see the following paragraph added in the result section
Severe EA – clinical significant dysphagia, clinically significant gastro – esophageal reflux disease, received dilatation of esophagus, airway disease (3 patients, for 2-7 years old and 8- 17 years old) ; moderate EA – clinically significant dysphagia, gastro – esophageal reflux disease, received dilatation of esophagus or clinically significant airway disease with associated anomaly (4 patients, for 2-7 years old and 8- 17 years old); mild EA – dysphagia or gastroesophageal reflux disease or airway disease, no associated anomaly (2 patients, for 2-7 years old and 8- 17 years old).
REVIEWER COMMENTS 3
Table 1: were there really 11 children (in age group 2–7 years) with multiple pregnancy (>45%) while only 1 child (3.7%) in older age group? Also, abbreviations are given for median (Me) and mean (M), but those are not used in the Table.
Thank you for this comment. Yes, it's true that these are the results we got.
Table 2 is very confusing, not clear at all. Also, what does ‘y’ stand for which is used after symptoms like heartburn etc.
Thank you for this comment. We have corrected the table to be more clear.
To summarize, the authors have done a great job with the translation and it appears to me that the questionnaire validation is carried out well. This is definitely relevant for the Polish population and especially families and children with esophageal atresia and the teams looking after them.
Thank you for your encouraging remarks.